# Identification of Yeasts with Mass Spectrometry during Wine Production

**Miroslava Kačániová** [1,2,*] , **Simona Kunová** [3] , **Jozef Sabo** [1] , **Eva Ivanišová** [4] , **Jana Žiarovská** [5] , **Soňa Felsöciová** [6] **and Margarita Terentjeva** [7]

1. Department of Fruit Sciences, Viticulture and Enology, Faculty of Horticulture and Landscape Engineering, Slovak University of Agriculture, Tr. A. Hlinku 2, 94976 Nitra, Slovakia; sabododik@gmail.com
2. Department of Bioenergy, Food Technology and Microbiology, Institute of Food Technology and Nutrition, University of Rzeszow, Zelwerowicza St. 4, 35601 Rzeszow, Poland
3. Department of Food Hygiene and Safety, Faculty of Biotechnology and Food Sciences, Slovak University of Agriculture, Tr. A. Hlinku 2, 94976 Nitra, Slovakia; simona.kunova@uniag.sk or simona.kunova@gmail.com
4. Department of Technology and Quality of Plant Products, Faculty of Biotechnology and Food Sciences, Slovak University of Agriculture, Tr. A. Hlinku 2, 94976 Nitra, Slovakia; eva.ivanisova@uniag.sk
5. Department of Plant Genetics and Breeding, Faculty of Agrobiology and Food Resources, Slovak University of Agriculture, Tr. A. Hlinku 2, 94976 Nitra, Slovakia; jana.ziarovska@uniag.sk
6. Department of Microbiology, Faculty of Biotechnology and Food Sciences, Slovak University of Agriculture, Tr. A. Hlinku 2, 94976 Nitra, Slovakia; sona.felsociova@uniag.sk
7. Institute of Food and Environmental Hygiene, Faculty of Veterinary Medicine, Latvia University of Life Sciences and Technologies, K. Helmaņa iela 8, LV-3004 Jelgava, Latvia; margarita.terentjeva@llu.lv
* Correspondence: miroslava.kacaniova@gmail.com; Tel.: +421-905-499-166

**Abstract:** The aim of the present study was to identify yeasts in grape, new wine "federweisser" and unfiltered wine samples. A total amount of 30 grapes, 30 new wine samples and 30 wine samples (15 white and 15 red) were collected from August until September, 2018, from a local Slovak winemaker, including Green Veltliner (3), Mūller Thurgau (3), Palava (3), Rhein Riesling (3), Sauvignon Blanc (3), Alibernet (3), André (3), Blue Frankish (3), Cabernet Sauvignon (3), and Dornfelder (3) grapes; federweisser and unfiltered wine samples were also used in our study. Wort agar (WA), yeast extract peptone dextrose agar (YPDA), malt extract agar (MEA) and Sabouraud dextrose agar (SDA) were used for microbiological testing of yeasts. MALDI-TOF Mass Spectrometry (Microflex LT/SH) (Bruker Daltonics, Germany) was used for the identification of yeasts. A total of 1668 isolates were identified with mass spectrometry. The most isolated species from the grapes was *Hanseniaspora uvarum*, and from federweisser and the wine—*Saccharomyces cerevisiae*.

**Keywords:** yeasts; grape; federweisser; wine; microbiota identification; MALDI-TOF MS Biotyper

## 1. Introduction

Yeasts naturally occur in wines and vineyards and are especially common on the grapes. Population of yeast species on the grape is not constant and increases during the ripening process. *Kloeckera apiculata* is a lemon-like cell shape yeast, which colonizes the grape surface [1]. *Kloeckera apiculata* comprises more than 50% of the total healthy grape microbiota. Other yeasts like *Kloeckera* were isolated from the surface of the grapes, which included mainly genera *Metschnikowia*, *Candida*, *Cryptococcus*, *Pichia*, *Rhodotorula*, *Zygosaccharomyces* or *Kluyveromyces* [2]. The presence of yeasts of the genus *Aureobasidium* attracted attention as a transitional genus between yeast and microscopic fungi. All the yeasts associated with natural microbiota of grapes are wild yeast strains or non-saccharomyces. Despite the presence of those yeasts on the surface of grapes, the wine production consists of subsequent fermentation stages,

which are typical for only particular yeast genera [3]. The *Saccharomyces* genus is the most important for the wine making process; however, this yeast is found on the grapes only in very small amounts. Previous studies that counted *Saccharomyces* on grapes found as little as 50 CFU/g. Mostly wild yeasts cultures could be found on the grapes and in freshly pressed must with colonization rates of $10^3$ to $10^5$ CFU/mL. During alcoholic fermentation, *Saccharomyces cerevisiae* is dominant, while yeasts in the *Pichia* and *Candida* genera are widespread in finished wine. The osmotolerant yeasts *Zygosaccharomyces* were reported in wines with higher content of residual sugar; yeasts of the *Brettanomyces* genus were common for wines in barrels [4,5].

The most important yeasts associated with wine production were Hanseniaspora uvarum (anamorph Kloeckera apiculata), Metschnikowia pulcherrima, Rhodotorula mucilaginosa, Rhodotorula glutinis, Aureobasidium pullulans, Cryptococcus magnus, Pichia manshurica, Pichia membranifaciens (anamorph Candida valida), Pichia fermentans, Pichia kluyveri, Pichia occidentalis (anamorph Candida sorbosa), Wickerhamomyces anomalus (anamorph Candida pelliculosa; Pichia anomala is synonymous), Cyberlindnera jadinii (Pichia jadinii is synonymous), Kregervanrija fluxuum (anamorph Candida vini), Candida stellata, Candida inconspicua, Meyerozyma guilliermondii, Zygosaccharomyces bailii, Brettanomyces bruxellensis (teleomorph Dekkera bruxellensis), Saccharomycodes ludwigii, Torulaspora delbrueckii and Saccharomyces cerevisiae. Kluyveromyces marxianus and Debaryomyces hansenii associated with grapes and are known as a contaminant in wine production. The microbiota of grapes creates better conditions for the growth of yeasts rather than bacteria. Low pH (pH 3–3.3), high content of sugars (mainly glucose) in grapes, and an anaerobic environment in must are necessary for ethanol fermentation of sugars, converting them into alcohol (ethanol) and $CO_2$ [5–8].

The aim of this study was to identify yeasts in grapes, federweisser and wine samples.

## 2. Materials and Methods

### 2.1. Collection of Grape, Federweisser and Wine Samples

An amount of 90 samples, including grape berries ($n = 30$), federweisser ($n = 30$) and wine ($n = 30$) of *Vitis vinifera* were collected aseptically in the viticultural area of Vrbové (approximately 48°37′12″ N and 017°43′25″ E) in 2018. The grape berry samples were transported on ice and stored at −20 °C until processing. The white grape varieties Green Veltliner, Mūller Thurgau, Palava, Rhein Riesling and Sauvignon Blanc as well as red grape varieties Alibernet, André, Blue Frankish, Cabernet Sauvignon and Dornfelder were collected. Three sampling points in distal spatial points of different rows were used for sampling of grape berries. Grape samples were collected in August, and processed independently.

Samples of "federweisser" were collected at the end of August 2018 and in the middle of September 2018 from the same winery as the grapes. Samples were collected into 200 mL sterile plastic bottles and stored at 8 ± 1 °C in a refrigerator. Before testing, the samples ($n = 30$) were diluted with sterile physiological saline (0.85%). A total of 100 μL of each dilution ($10^{-1}$ to $10^{-5}$) was used for microbiological testing.

An amount of 200 mL of each unfiltered wine (before microfiltration) and immediately after were stored at 6–8 °C in a refrigerator. Collected wine samples were fermented with *Saccharomyces cerevisiae* in the producing process. The samples were later incubated in the laboratory at room temperature (25 ± 2 °C) for one week until the laboratory testing was initiated.

### 2.2. Cultivation Media

Wort agar (WA) (HiMedia, Mumbai, India), yeast extract peptone dextrose agar (YPDA) (Conda, Madrid, Spain), malt extract agar (MEA) (Biomark, Maharashtra, India) and Sabouraud dextrose agar (SDA) (Conda, Madrid, Spain) were used for identification of yeasts. All media were supplemented with chloramphenicol (100 mg/L) to inhibit bacterial growth. Chloramphenicol (Biolife, Monza, Italy) was added into cultivation media before sterilization by autoclaving at 115–121 °C for 15 min. The acid base indicator bromocresol green (BG, Biolofe, Monza, Italy) (20 mg/L) (pH range: 3.8–5.4) was

added into the MEA and WA cultivation media before sterilization. Media for yeast cultivation were inoculated with 100 μL of the sample suspension. Inoculated agars were incubated at 25 °C for 3–5 days and the yeasts were identified by colony morphology (colour, surface, edge and elevation) and reinoculated onto trypton soya agar (TSA) (Oxoid, Basingstoke, UK). Yeast species were identified with a MALDI-TOF MS Biotyper.

## 2.3. Identification of Isolates with Mass Spectrometry

Qualitative analysis of yeasts isolates was performed with MALDI-TOF mass spectrometry (Bruker Daltonics, Bremen, Germany). Isolates were put in 300 μL of distilled water and 900 μL of ethanol, and the suspension centrifuged for 2 min at 14,000 rpm. The pellet was centrifuged repeatedly and allowed to dry. An amount of 30 μL of 70% formic acid was added to the pellet and 30 μL of acetonitrile. Tubes were centrifuged for 2 min at 14,000 rpm and 1 μL of the supernatant was used for MALDI identification. Once dry, every spot was overlaid with 1 μL of an HCCA matrix and left to dry at room temperature before analysis. Generated spectra were analyzed on a MALDI-TOF Microflex LT (Bruker Daltonics, Bremen, Germany) instrument using Flex Control 3.4 software and Biotyper Realtime Classification 3.1 with BC-specific software. Criteria for reliable identification were a score of ≥2.0 at species level [9].

## 2.4. Statistical Analysis

The statistical processing of the data obtained from each evaluation was done with Statgraphics Plus version 5.1 (AV Trading, Umex, Dresden, Germany). For each replication the mean was calculated, and the data set were log transformed. Descriptive statistics and logical-cognitive methods and one-way analysis ANOVA were used in the evaluation and statistical analysis.

## 3. Results and Discussion

Grapes are inhabited by versatile microbial groups and have a complex microbial ecology, including filamentous fungi, yeasts and bacteria. These microorganisms pose different physiological characteristics and may affect the wine quality. Some species of parasitic fungi or environmental bacteria might be only found in grapes, while other microorganisms like yeast, lactic acid and acetic acid bacteria occur during the winemaking process [10].

The yeast count in grape ranged from 2.34 (Greener Veltliner) to 2.67 (Dornfelder) log CFU/g on MEA, from 2.19 (Mūller Thurgau) to 2.38 (Dornfelder) log CFU/g on WA, from 2.46 (Greener Veltliner) to 2.66 (Dornfelder) log CFU/g on YPDA, and from 1.55 (Greener Veltliner) to 1.88 (Dornfelder) log CFU/g on SDA. The colonization of grapes with yeasts is shown in Table 1.

**Table 1.** Yeasts counts in grape berries on different media.

| Sample | MEA | WA | YPDA | SDA |
|---|---|---|---|---|
| | Microbial Counts log CFU/g | | | |
| Green Veltliner | 2.37 ± 0.14 | 2.22 ± 0.06 | 2.46 ± 0.05 | 1.55 ± 0.14 |
| Mūller Thurgau | 2.34 ± 0.09 | 2.19 ± 0.04 | 2.49 ± 0.04 | 1.66 ± 0.22 |
| Palava | 2.36 ± 0.13 | 2.20 ± 0.07 | 2.52 ± 0.01 | 1.69 ± 0.17 |
| Rhein Riesling | 2.43 ± 0.01 | 2.17 ± 0.06 | 2.51 ± 0.02 | 1.67 ± 0.16 |
| Sauvignon Blanc | 2.40 ± 0.03 | 2.19 ± 0.04 | 2.49 ± 0.02 | 1.65 ± 0.12 |
| Alibernet | 2.64 ± 0.10 | 2.26 ± 0.08 | 2.53 ± 0.03 | 1.73 ± 0.16 |
| André | 2.66 ± 0.07 | 2.33 ± 0.10 | 2.57 ± 0.06 | 1.79 ± 0.05 |
| Blue Frankish | 2.64 ± 0.03 | 2.36 ± 0.06 | 2.59 ± 0.06 | 1.82 ± 0.04 |
| Cabernet Sauvignon | 2.66 ± 0.03 | 2.34 ± 0.09 | 2.64 ± 0.01 | 1.84 ± 0.02 |
| Dornfelder | 2.67 ± 0.05 | 2.38 ± 0.04 | 2.66 ± 0.04 | 1.88 ± 0.06 |

WA—wort agar; YPDA—yeast extract peptone dextrose agar; MEA—malt extract agar; SDA—Sabouraud dextrose agar.

ANOVA analysis was performed to inspect the significant differences among the microbial count for individual wine varieties when different cultivation media were used (Table 2).

**Table 2.** One-way ANOVA for analyzed wine varieties—grapes.

| Cultivation Media | Source | Sum of Squares | Degrees of Freedom | Mean Square | F Statistic | *p*-Value |
|---|---|---|---|---|---|---|
| MEA | treatment | 0.5779 | 9 | 0.0642 | 9.55 | $1.65 \times 10^{-5}$ |
| | error | 0.1347 | 20 | 0.0067 | | |
| | total | 0.7125 | 29 | | | |
| WA | treatment | 0.1755 | 9 | 0.0195 | 4.26 | 0.0025 |
| | error | 0.0868 | 20 | 0.0043 | | |
| | total | 0.2623 | 29 | | | |
| YPDA | treatment | 0.1179 | 9 | 0.0131 | 8.74 | $3.77 \times 10^{-5}$ |
| | error | 0.0307 | 20 | 0.0015 | | |
| | total | 0.1487 | 29 | | | |
| SDA | treatment | 0.2863 | 9 | 0.0318 | 1.13 | 0.1287 |
| | error | 0.3513 | 20 | 0.0176 | | |
| | total | 0.6376 | 29 | | | |

WA—wort agar; YPDA—yeast extract peptone dextrose agar; MEA—malt extract agar; SDA—Sabouraud dextrose agar.

Statistically significant differences among microbial counts for individual cultivation media were found in three of the four cultivation media used (Table 3).

**Table 3.** Significant differences among analyzed grape varieties for individual cultivation media.

| Treatments Pair | Tukey HSD *p*-Value | Tukey HSD Inferfence |
|---|---|---|
| **MEA** | | |
| A vs. F | 0.0180304 | * $p < 0.05$ |
| A vs. G | 0.0085086 | ** $p < 0.01$ |
| A vs. H | 0.0180304 | * $p < 0.05$ |
| A vs. I | 0.0076352 | ** $p < 0.01$ |
| A vs. J | 0.0068497 | ** $p < 0.01$ |
| B vs. F | 0.0085086 | ** $p < 0.01$ |
| B vs. G | 0.0039776 | ** $p < 0.01$ |
| B vs. H | 0.0085086 | ** $p < 0.01$ |
| B vs. I | 0.0035659 | ** $p < 0.01$ |
| B vs. J | 0.0032002 | ** $p < 0.01$ |
| C vs. F | 0.0130929 | * $p < 0.05$ |
| C vs. G | 0.0061450 | ** $p < 0.01$ |
| C vs. H | 0.0130929 | * $p < 0.05$ |
| C vs. I | 0.0055127 | ** $p < 0.01$ |
| C vs. J | 0.0049454 | ** $p < 0.01$ |
| E vs. G | 0.0247440 | * $p < 0.05$ |
| E vs. I | 0.0222763 | * $p < 0.05$ |
| E vs. J | 0.0200452 | * $p < 0.05$ |
| **WA** | | |
| D vs. H | 0.0395384 | * $p < 0.05$ |
| D vs. J | 0.0206542 | * $p < 0.05$ |
| **YPDA** | | |
| A vs. H | 0.0262122 | * $p < 0.05$ |
| A vs. I | 0.0010053 | ** $p < 0.01$ |
| A vs. J | 0.0010053 | ** $p < 0.01$ |
| B vs. I | 0.0068426 | ** $p < 0.01$ |

**Table 3.** *Cont.*

| Treatments Pair | Tukey HSD *p*-Value | Tukey HSD Inference |
|---|---|---|
| YPDA | | |
| B vs. J | 0.0011070 | ** *p* < 0.01 |
| C vs. J | 0.0085840 | ** *p* < 0.01 |
| D vs. I | 0.0210350 | * *p* < 0.05 |
| D vs. J | 0.0034566 | ** *p* < 0.01 |
| E vs. I | 0.0043409 | ** *p* < 0.01 |
| E vs. J | 0.0010053 | ** *p* < 0.01 |
| F vs. J | 0.0168484 | * *p* < 0.05 |

A—Green Veltliner; B—Müller Thurgau; C—Palava; D—Rhein Riesling; E—Sauvignon Blanc; F—Alibernet; G—André; H—Blue Frankish; I—Cabernet Sauvignon; J—Dornfelder; WA—wort agar; YPDA—yeast extract peptone dextrose agar; MEA—malt extract agar; SDA—Sabouraud dextrose agar.

Different studies have evaluated the surface microbiota of grape berries due to a possible impact on the hygienic state of the grapes and the direct influence on the winemaking process and wine quality [11–18].

The yeasts count in "federweisser" ranged from 3.51 in Greener Veltliner and Palava to 3.80 log CFU/mL in Dornfelder on MEA. On WA, the yeasts count from 3.30 in Palava to 3.53 log CFU/mL in Dornfelder were observed. On YPDA, the yeasts count varied from 3.24 in Rhein Riesling to 3.45 log CFU/mL in Dornfelder, and from 3.13 (Sauvignon Blanc) to 3.33 (Dornfelder) log CFU/mL on SDA. Yeasts counts in federweisser are summarized in Table 4.

**Table 4.** Yeast counts in "federweisser" on different media.

| Sample | MEA | WA | YPDA | SDA |
|---|---|---|---|---|
| | log CFU/g | | | |
| Green Veltliner | 3.51 ± 0.15 | 3.41 ± 0.06 | 3.37 ± 0.14 | 3.21 ± 0.01 |
| Müller Thurgau | 3.54 ± 0.10 | 3.38 ± 0.06 | 3.30 ± 0.04 | 3.19 ± 0.03 |
| Palava | 3.51 ± 0.05 | 3.30 ± 0.06 | 3.27 ± 0.04 | 3.16 ± 0.05 |
| Rhein Riesling | 3.58 ± 0.06 | 3.34 ± 0.01 | 3.24 ± 0.01 | 3.14 ± 0.02 |
| Sauvignon Blanc | 3.56 ± 0.10 | 3.36 ± 0.05 | 3.27 ± 0.05 | 3.13 ± 0.02 |
| Alibernet | 3.67 ± 0.08 | 3.40 ± 0.06 | 3.29 ± 0.06 | 3.17 ± 0.03 |
| André | 3.70 ± 0.07 | 3.43 ± 0.02 | 3.36 ± 0.05 | 3.18 ± 0.06 |
| Blue Frankish | 3.74 ± 0.02 | 3.46 ± 0.05 | 3.39 ± 0.05 | 3.24 ± 0.09 |
| Cabernet Sauvignon | 3.76 ± 0.05 | 3.48 ± 0.06 | 3.41 ± 0.01 | 3.30 ± 0.04 |
| Dornfelder | 3.80 ± 0.07 | 3.53 ± 0.03 | 3.45 ± 0.06 | 3.33 ± 0.01 |

WA—wort agar; YPDA—yeast extract peptone dextrose agar; MEA—malt extract agar; SDA—Sabouraud dextrose agar.

In study in Slovakia [19], the highest yeasts counts were on MEA for Pinot Noir—6.43 log CFU/mL and the lowest for Moravian Muscat—4.62 log CFU/mL. The highest yeasts count on WA were in Pinot Noir—6.39 log CFU/mL, but the lowest in Irsai Oliver—5.38 log CFU/mL. The highest count of yeasts on wild yeast medium (WYM) was in Blue Frankish 6.33 log CFU/mL and the lowest in Dornfelder 4.20 log CFU/mL [19].

As the results show, a higher number of yeasts were detected in "federweisser" than in grape. The young wine is a product of fermentation where *S. cerevisiae* was mostly found. Other species like *Hanseniaspora uvarum*, *Metschnikowia pulcherrima* or the genera *Pichia* or *Candida* may be present during the individual fermentation stages when the alcohol content do not exceed 4–6% [5,20]. The main microbiota of the grape is the yeast *Hanseniaspora uvarum* followed by *Metschnikowia pulcherrima* [4]. These species also initiate the pre-alcoholic fermentation but are being replaced by the dominant *S. cerevisiae* 3–4 days after fermentation. *Saccharomyces cerevisiae* starts to multiply within 20 days after inoculation into the must [21].

ANOVA analysis was performed to inspect the significant differences among the microbial count for individual wine varieties when different cultivation media were used (Table 5).

**Table 5.** One-way ANOVA results for the analyzed wine varieties—federweisser.

| Cultivation Media | Source | Sum of Squares | Degrees of Freedom | Mean Square | F Statistic | *p*-Value |
|---|---|---|---|---|---|---|
| MEA | treatment | 0.3717 | 9 | 0.0413 | 6.36 | 0.0002 |
| | error | 0.1234 | 20 | 0.0062 | | |
| | total | 0.4951 | 29 | | | |
| WA | treatment | 0.1305 | 9 | 0.0145 | 5.23 | 0.0007 |
| | error | 0.0525 | 20 | 0.0026 | | |
| | total | 0.1831 | 29 | | | |
| YPDA | treatment | 0.1426 | 9 | 0.0158 | 3.36 | 0.0056 |
| | error | 0.0818 | 20 | 0.0041 | | |
| | total | 0.2244 | 29 | | | |
| SDA | treatment | 0.1194 | 9 | 0.0133 | 6.91 | 0.0002 |
| | error | 0.0397 | 20 | 0.0020 | | |
| | total | 0.1591 | 29 | | | |

WA—wort agar; YPDA—yeast extract peptone dextrose agar; MEA—malt extract agar; SDA—Sabouraud dextrose agar.

Statistically significant differences among microbial counts for individual cultivation media were found in three of the four cultivation media used (Table 6).

**Table 6.** Significant differences among analyzed federweisser samples for individual cultivation media.

| Reatments Pair | Tukey HSD *p*-Value | Tukey HSD Inferfence |
|---|---|---|
| MEA | | |
| A vs. H | 0.0368638 | * $p < 0.05$ |
| A vs. I | 0.0238533 | * $p < 0.05$ |
| A vs. J | 0.0055561 | ** $p < 0.01$ |
| B vs. H | 0.0456379 | * $p < 0.05$ |
| B vs. I | 0.0296858 | * $p < 0.05$ |
| B vs. J | 0.0069685 | ** $p < 0.01$ |
| C vs. H | 0.0410343 | * $p < 0.05$ |
| C vs. I | 0.0266151 | * $p < 0.05$ |
| C vs. J | 0.0062207 | ** $p < 0.01$ |
| E vs. H | 0.0456379 | * $p < 0.05$ |
| E vs. I | 0.0296858 | * $p < 0.05$ |
| E vs. J | 0.0069685 | ** $p < 0.01$ |
| WA | | |
| C vs. H | 0.0329699 | * $p < 0.05$ |
| C vs. I | 0.0100494 | * $p < 0.05$ |
| C vs. J | 0.0010463 | ** $p < 0.01$ |
| D vs. J | 0.0084520 | ** $p < 0.01$ |
| E vs. J | 0.0071077 | ** $p < 0.01$ |
| YPDA | | |
| D vs. J | 0.0105986 | * $p < 0.05$ |
| E vs. J | 0.0359336 | * $p < 0.05$ |
| SDA | | |
| B vs. J | 0.0319805 | * $p < 0.05$ |
| C vs. I | 0.0217427 | * $p < 0.05$ |
| C vs. J | 0.0044497 | ** $p < 0.01$ |

**Table 6.** *Cont*.

| Reatments Pair | Tukey HSD *p*-Value | Tukey HSD Inferfence |
|---|---|---|
| SDA | | |
| D vs. I | 0.0081105 | ** *p* < 0.01 |
| D vs. J | 0.0016306 | ** *p* < 0.01 |
| E vs. I | 0.0036406 | ** *p* < 0.01 |
| E vs. J | 0.0010053 | ** *p* < 0.01 |
| F vs. I | 0.0466723 | * *p* < 0.05 |
| F vs. J | 0.0098975 | ** *p* < 0.01 |
| G vs. J | 0.0178947 | * *p* < 0.05 |

A—Green Veltliner; B—Müller Thurgau; C—Palava; D—Rhein Riesling; E—Sauvignon Blanc; F—Alibernet; G—André; H—Blue Frankish; I—Cabernet Sauvignon; J—Dornfelder; WA—wort agar; YPDA—yeast extract peptone dextrose agar; MEA—malt extract agar; SDA—Sabouraud dextrose agar.

The yeast counts in the unfiltered wines are summarized in Table 7. The yeast counts in wine ranged from 1.51 (Greener Veltliner) to 3.23 (Dornfelder) log CFU/mL on MEA, from 1.43 (Greener Veltliner) to 2.89 (Dornfelder) log CFU/mL on WA, from 1.18 (Greener Veltliner) to 2.65 (Dornfelder) log CFU/mL on YPDA and from 1.09 (Rhein Riesling) to 2.21 (Dornfelder) log CFU/mL on SDA.

**Table 7.** Yeast counts in wine on different media.

| Sample | MEA | WA | YPDA | SDA |
|---|---|---|---|---|
| | log CFU/g | | | |
| Green Veltliner | 1.51 ± 0.27 | 1.43 ± 0.15 | 1.18 ± 0.06 | 1.13 ± 0.03 |
| Müller Thurgau | 1.55 ± 0.32 | 1.52 ± 0.01 | 1.21 ± 0.06 | 1.18 ± 0.06 |
| Palava | 1.62 ± 0.34 | 1.49 ± 0.03 | 1.27 ± 0.11 | 1.12 ± 0.02 |
| Rhein Riesling | 1.73 ± 0.17 | 1.46 ± 0.05 | 1.27 ± 0.14 | 1.09 ± 0.06 |
| Sauvignon Blanc | 1.76 ± 0.11 | 1.48 ± 0.06 | 1.32 ± 0.23 | 1.14 ± 0.02 |
| Alibernet | 2.57 ± 0.50 | 2.37 ± 0.14 | 2.21 ± 0.05 | 1.51 ± 0.64 |
| André | 2.67 ± 0.42 | 2.41 ± 0.10 | 2.31 ± 0.13 | 1.83 ± 0.62 |
| Blue Frankish | 2.59 ± 0.28 | 2.44 ± 0.16 | 2.38 ± 0.11 | 2.13 ± 0.12 |
| Cabernet Sauvignon | 2.89 ± 0.37 | 2.51 ± 0.14 | 2.47 ± 0.04 | 2.17 ± 0.07 |
| Dornfelder | 3.23 ± 0.02 | 2.89 ± 0.27 | 2.65 ± 0.22 | 2.21 ± 0.06 |

WA—wort agar; YPDA—yeast extract peptone dextrose agar; MEA—malt extract agar; SDA—Sabouraud dextrose agar.

ANOVA analysis was performed to inspect the significant differences among the microbial count for individual wine varieties when different cultivation media were used (Table 8).

**Table 8.** One-way ANOVA results for analyzed wine varieties—unfiltered wine.

| Cultivation Media | Source | Sum of Squares | Degrees of Freedom | Mean Square | F Statistic | *p*-Value |
|---|---|---|---|---|---|---|
| MEA | treatment | 11.0908 | 9 | 1.23 | 12.60 | $1.95 \times 10^{-6}$ |
| | error | 1.45 | 20 | 0.0982 | | |
| | total | 13.0552 | 29 | | | |
| WA | treatment | 8.05 | 9 | 0.9745 | 54.8807 | $3.84 \times 10^{-12}$ |
| | error | 0.3551 | 20 | 0.0178 | | |
| | total | 9.1256 | 29 | | | |
| YPDA | treatment | 10.74 | 9 | 1.1542 | 65.9142 | $6.70 \times 10^{-13}$ |
| | error | 0.3502 | 20 | 0.0175 | | |
| | total | 10.76 | 29 | | | |
| SDA | treatment | 6.61 | 9 | 0.7118 | 8.13 | $3.51 \times 10^{-5}$ |
| | error | 1.31 | 20 | 0.0827 | | |
| | total | 8.0592 | 29 | | | |

WA—wort agar; YPDA—yeast extract peptone dextrose agar; MEA—malt extract agar; SDA—Sabouraud dextrose agar.

Statistically significant differences among microbial count for individual cultivation media were found in three of the four cultivation media used (Table 9).

**Table 9.** Significant differences among unfiltered wine samples for individual cultivation media.

| Treatments Pair | Tukey HSD $p$-Value | Tukey HSD Inferfence |
|---|---|---|
| MEA | | |
| A vs. F | 0.0141715 | * $p < 0.05$ |
| A vs. G | 0.0064264 | ** $p < 0.01$ |
| A vs. H | 0.0119733 | * $p < 0.05$ |
| A vs. I | 0.0010053 | ** $p < 0.01$ |
| A vs. J | 0.0010053 | ** $p < 0.01$ |
| B vs. F | 0.0192566 | * $p < 0.05$ |
| B vs. G | 0.0087792 | ** $p < 0.01$ |
| B vs. H | 0.0162964 | * $p < 0.05$ |
| B vs. I | 0.0012652 | ** $p < 0.01$ |
| B vs. J | 0.0010053 | ** $p < 0.01$ |
| C vs. F | 0.0342577 | * $p < 0.05$ |
| C vs. G | 0.0158481 | * $p < 0.05$ |
| C vs. H | 0.0290959 | * $p < 0.05$ |
| C vs. I | 0.0023022 | ** $p < 0.01$ |
| C vs. J | 0.0010053 | ** $p < 0.01$ |
| D vs. G | 0.0381648 | * $p < 0.05$ |
| D vs. I | 0.0057354 | ** $p < 0.01$ |
| D vs. J | 0.0010053 | ** $p < 0.01$ |
| E vs. I | 0.0078360 | ** $p < 0.01$ |
| E vs. J | 0.0010053 | ** $p < 0.01$ |
| WA | | |
| A vs. F | 0.0010053 | ** $p < 0.01$ |
| A vs. G | 0.0010053 | ** $p < 0.01$ |
| A vs. H | 0.0010053 | ** $p < 0.01$ |
| A vs. I | 0.0010053 | ** $p < 0.01$ |
| A vs. J | 0.0010053 | ** $p < 0.01$ |
| B vs. F | 0.0010053 | ** $p < 0.01$ |
| B vs. G | 0.0010053 | ** $p < 0.01$ |
| B vs. H | 0.0010053 | ** $p < 0.01$ |
| B vs. I | 0.0010053 | ** $p < 0.01$ |
| B vs. J | 0.0010053 | ** $p < 0.01$ |
| C vs. F | 0.0010053 | ** $p < 0.01$ |
| C vs. G | 0.0010053 | ** $p < 0.01$ |
| C vs. H | 0.0010053 | ** $p < 0.01$ |
| C vs. I | 0.0010053 | ** $p < 0.01$ |
| C vs. J | 0.0010053 | ** $p < 0.01$ |
| D vs. F | 0.0010053 | ** $p < 0.01$ |
| D vs. G | 0.0010053 | ** $p < 0.01$ |
| D vs. H | 0.0010053 | ** $p < 0.01$ |
| D vs. I | 0.0010053 | ** $p < 0.01$ |
| D vs. J | 0.0010053 | ** $p < 0.01$ |
| E vs. F | 0.0010053 | ** $p < 0.01$ |
| E vs. G | 0.0010053 | ** $p < 0.01$ |
| E vs. H | 0.0010053 | ** $p < 0.01$ |
| E vs. I | 0.0010053 | ** $p < 0.01$ |
| E vs. J | 0.0010053 | ** $p < 0.01$ |
| F vs. J | 0.0037886 | ** $p < 0.01$ |
| G vs. J | 0.0079079 | ** $p < 0.01$ |
| H vs. J | 0.0143649 | * $p < 0.05$ |

**Table 9.** *Cont.*

| Treatments Pair | Tukey HSD *p*-Value | Tukey HSD Inferfence |
|---|---|---|
| YPDA | | |
| A vs. F | 0.0010053 | ** *p* < 0.01 |
| A vs. G | 0.0010053 | ** *p* < 0.01 |
| A vs. H | 0.0010053 | ** *p* < 0.01 |
| A vs. I | 0.0010053 | ** *p* < 0.01 |
| A vs. J | 0.0010053 | ** *p* < 0.01 |
| B vs. F | 0.0010053 | ** *p* < 0.01 |
| B vs. G | 0.0010053 | ** *p* < 0.01 |
| B vs. H | 0.0010053 | ** *p* < 0.01 |
| B vs. I | 0.0010053 | ** *p* < 0.01 |
| B vs. J | 0.0010053 | ** *p* < 0.01 |
| C vs. F | 0.0010053 | ** *p* < 0.01 |
| C vs. G | 0.0010053 | ** *p* < 0.01 |
| C vs. H | 0.0010053 | ** *p* < 0.01 |
| C vs. I | 0.0010053 | ** *p* < 0.01 |
| C vs. J | 0.0010053 | ** *p* < 0.01 |
| D vs. F | 0.0010053 | ** *p* < 0.01 |
| D vs. G | 0.0010053 | ** *p* < 0.01 |
| D vs. H | 0.0010053 | ** *p* < 0.01 |
| D vs. I | 0.0010053 | ** *p* < 0.01 |
| D vs. J | 0.0010053 | ** *p* < 0.01 |
| E vs. F | 0.0010053 | ** *p* < 0.01 |
| E vs. G | 0.0010053 | ** *p* < 0.01 |
| E vs. H | 0.0010053 | ** *p* < 0.01 |
| E vs. I | 0.0010053 | ** *p* < 0.01 |
| E vs. J | 0.0010053 | ** *p* < 0.01 |
| F vs. J | 0.0154088 | * *p* < 0.05 |
| SDA | | |
| A vs. H | 0.0103393 | * *p* < 0.05 |
| A vs. I | 0.0071377 | ** *p* < 0.01 |
| A vs. J | 0.0044827 | ** *p* < 0.01 |
| B vs. H | 0.0168657 | * *p* < 0.05 |
| B vs. I | 0.0116918 | * *p* < 0.05 |
| B vs. J | 0.0073639 | ** *p* < 0.01 |
| C vs. H | 0.0100244 | * *p* < 0.05 |
| C vs. I | 0.0069202 | ** *p* < 0.01 |
| C vs. J | 0.0043452 | ** *p* < 0.01 |
| D vs. H | 0.0075929 | ** *p* < 0.01 |
| D vs. I | 0.0052344 | ** *p* < 0.01 |
| D vs. J | 0.0032871 | ** *p* < 0.01 |
| E vs. H | 0.0116918 | * *p* < 0.05 |
| E vs. I | 0.0080777 | ** *p* < 0.01 |
| E vs. J | 0.0050746 | ** *p* < 0.01 |

A—Green Veltliner; B—Müller Thurgau; C—Palava; D—Rhein Riesling; E—Sauvignon Blanc; F—Alibernet; G—André; H—Blue Frankish; I—Cabernet Sauvignon; J—Dornfelder; WA—wort agar; YPDA—yeast extract peptone dextrose agar; MEA—malt extract agar; SDA—Sabouraud dextrose agar.

Altogether, 1668 isolates were identified with mass spectrometry with a score of ≥2.0 (Table 10). The most isolated species from grape was *Hanseniaspora uvarum* (70 isolates), and from "federweisser" and wine *S. cerevisiae* (85 and 120 isolates, respectively). Yeasts species of grape, "frederweisser" and wine are shown in Figures 1–3.

**Table 10.** Yeasts species in grape, "federweisser" and wine.

| Yeast Species | Grape | "Federweisser" No. of Isolates | Wine |
|---|---|---|---|
| *Aureobasidium pullulans* | 25 | 0 | 0 |
| *Candida inconspicua* | 5 | 0 | 5 |
| *Candida parapsilosis* | 5 | 0 | 10 |
| *Candida saitoana* | 5 | 0 | 5 |
| *Candida sake* | 5 | 0 | 5 |
| *Cyberlindnera jadinii* | 0 | 0 | 8 |
| *Debaryomyces hansenii* | 0 | 0 | 15 |
| *Dekkera bruxellensis* | 0 | 0 | 25 |
| *Filobasidium magnum* | 30 | 0 | 0 |
| *Hanseniaspora uvarum* | 70 | 25 | 0 |
| *Issatchenkia orientalis* | 38 | 0 | 0 |
| *Kazachstania exigua* | 33 | 0 | 0 |
| *Kluyveromyces marxianus* | 35 | 0 | 32 |
| *Kregervanrija fluxuum* | 0 | 0 | 25 |
| *Metschnikowia pulcherrima* | 28 | 50 | 55 |
| *Meyerozyma guilliermondii* | 0 | 0 | 52 |
| *Naganishia diffluens* | 25 | 0 | 0 |
| *Pichia fermentans* | 10 | 0 | 58 |
| *Pichia kluyveri* | 12 | 49 | 50 |
| *Pichia mandshurica* | 10 | 0 | 25 |
| *Pichia membranifaciens* | 25 | 0 | 15 |
| *Pichia norvegensis* | 10 | 0 | 5 |
| *Pichia occidentalis* | 10 | 35 | 45 |
| *Rhodotorula glutinis* | 40 | 0 | 20 |
| *Rhodotorula mucilaginosa* | 25 | 0 | 45 |
| *Saccharomyces cerevisiae* | 0 | 85 | 120 |
| *Starmerella magnolia* | 30 | 0 | 15 |
| *Torulaspora delbrueckii* | 12 | 0 | 39 |
| *Wickerhamomyces anomalus* | 15 | 0 | 75 |
| *Yarrowia lipolytica* | 20 | 0 | 10 |
| *Zygosaccharomyces bailii* | 15 | 0 | 52 |
| *Zygotorulaspora florentina* | 25 | 0 | 50 |
| **Total** | 563 | 244 | 861 |

*Brettanomyces bruxellensis*, *Candida stellata*, *Saccharomyces cerevisiae* and *Zygosaccharomyces bailii* were the yeasts identified in wine [22–25]. In our study, *Pichia mandshurica*—the main contaminant of wines—was present in 66% samples of white wines (10 out of 15) and in seven samples of red wines (46%). *Pichia membranifaciens* was isolated from five samples of white (33%) and five samples of red wines (33%). *Saccharomyces cerevisiae* was isolated from all white and red wines (100%). *Zygosaccharomyces bailii* was found in 14 samples of white (93%) and two samples of red (13%) wines. Our study shows that *Z. bailii* and *P. mandshurica* were isolated more frequently from white than from red wines, while *S. cerevisiae* was identified in white and red wines. The occurrence of *Pichia manshurica* and *S. cerevisiae* was different between the wine samples. According to Thomas [26], the presence of *Zygosaccharomyces* in wine is unacceptable in terms of wine quality. The author has stated that the minimum number of yeast present in wine spoils the product under appropriate conditions [26]. *Saccharomyces cerevisiae*, *Debaryomyces hansenii*, *Wickerhamomyces anomalus (Pichia anomala)*, *Pichia membranifaciens*, *Rhodotorula glutinis*, *Rhodotorula mucilaginosa*, *Torulaspora delbrueckii*, *Kluyveromyces marxianus*, *Issatchenkia orientalis*, *Zygosaccharomyces bailii parapsilosis*, *Pichia fermentans* and *Hanseniaspora uvarum* are frequent contaminants of wines as well [27,28]. However, Renous [29] did not describe associations between wine and *Pichia manshurica*, *Kregervanrija fluxuum (Candida vini)*, *Candida inconspicua* and *Zygotorulaspora florentina*. Saez [30] found that *S. cerevisiae* (13.93%),

*Wickerhamomyces anomalus* (8.72%), *Pichia fermentans* (6.74%) and *Metschnikowia pulcherrima* (6.39%) were the most abundant in wine. *Pichia* (*Pichia manshurica*, *P. membranifaciens*) and *Brettanomyces* are producing volatile phenols, thereby affecting the quality of the wine [30].

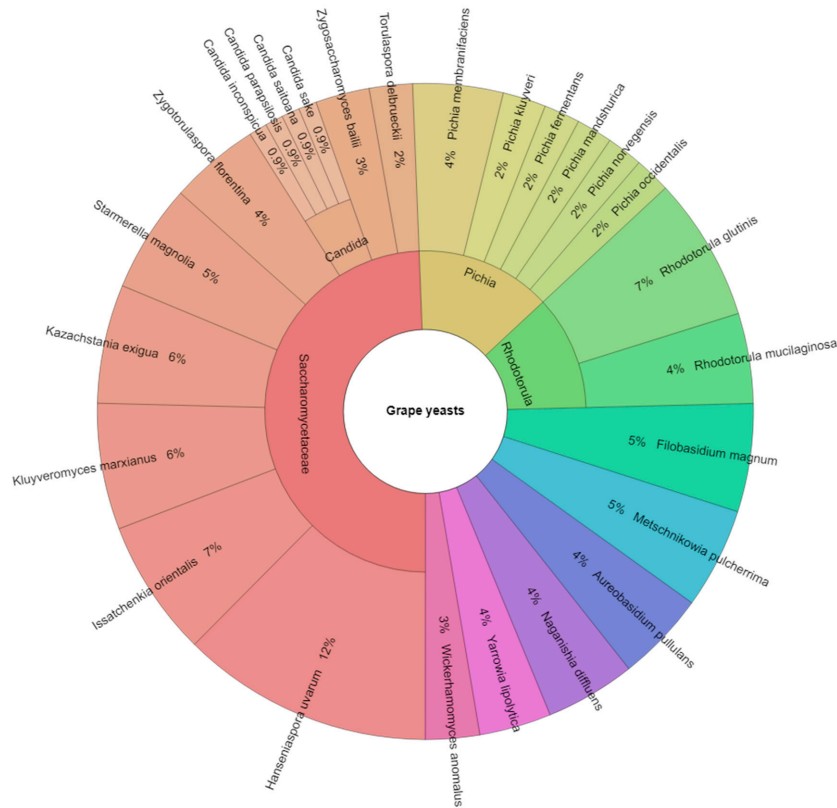

**Figure 1.** Yeasts isolated from the grapes.

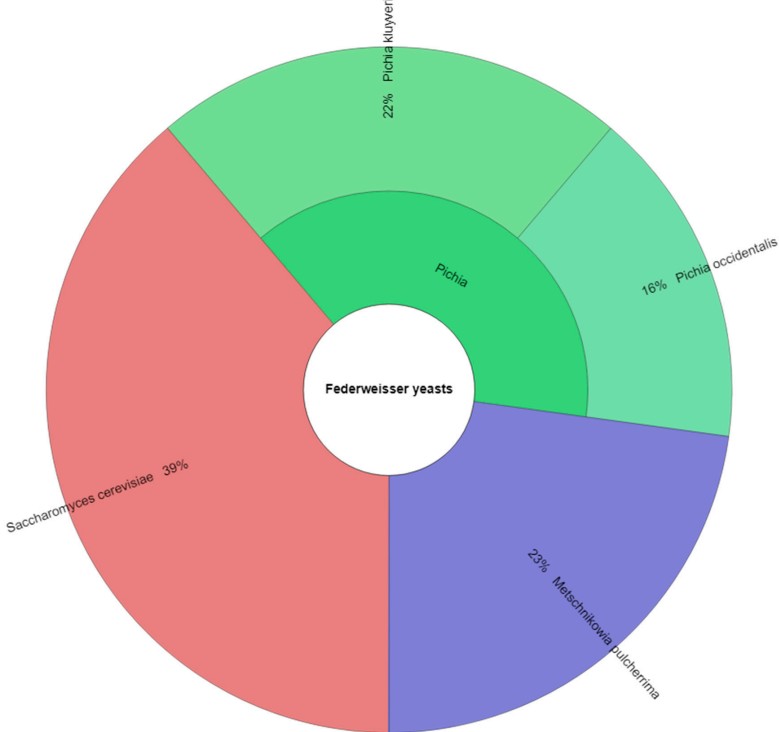

**Figure 2.** Yeasts isolated from the "federweisser".

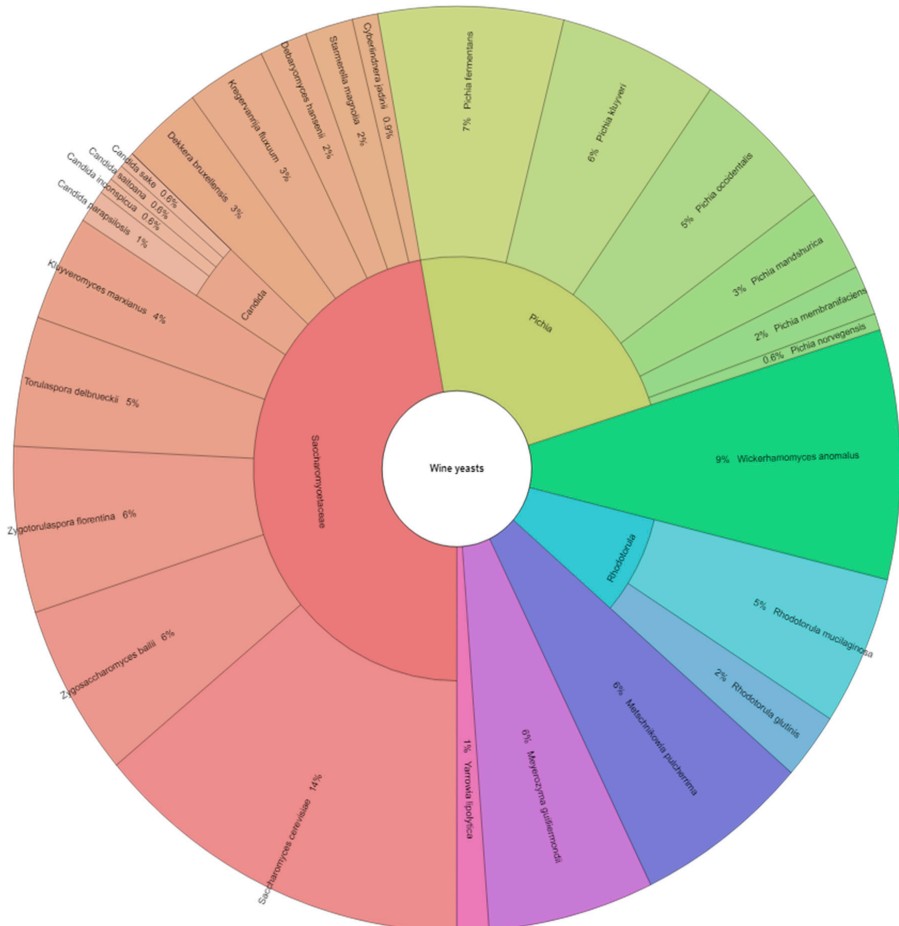

**Figure 3.** Yeasts isolated from the wine.

Sporadically, *Candida inconspicua* (5 isolates, 0.58%), *Candida saitoana* (5 isolates, 0.58%), *Candida sake* (5 isolates, 0.58%), *Pichia norvegensis* (5 isolates, 0.58%) and other species were isolated. Jolly et al. [31] noticed the importance of *Candida*, *Cryptococcus*, *Kloeckera* and *Rhodotorula* species in the wine making process. *Candida* was considered as the dominant genus, including their teleomorphic stages—*Candida pulcherrima* (*Metschnikowia pulcherrima*), *Candida vini* (*Kregervanrija fluxuum*) and *Candida valida* (*Pichia membranifaciens*) [31].

## 4. Conclusions

A total of 90 samples (30 from grapes, 30 of "federweisser" and 30 of wine) was studied for characterization of the yeast species. The mass spectrometry method was used for identification of 1668 grape, "federweisser" and wine isolates. From grape, 26 species of 17 genera within 9 families, and in "federweisser" 4 species of 3 genera and families were found. In wine, 26 species of 17 genera within 6 families were identified. *Rhodoturulla* species were not included in any family and they were classified as incertae sedis (not belonging anywhere).

**Author Contributions:** M.K., M.T., J.Ž. were responsible for the design of the study; M.K., S.K., J.S., S.F., conducted the study and collected the samples; M.K., S.K., J.S., J.Ž. performed the laboratory analysis; M.K., S.K., E.I., M.T. were responsible for writing and editing the manuscript; all authors have carefully revised and approved the final version of the manuscript. All authors have read and agreed to the published version of the manuscript.

**Funding:** This work has been supported by the grants of the Slovak Research and Development Agency No. VEGA 1/0411/17.

**Acknowledgments:** The Paper was supported by the project: The research leading to these results has received funding from the European Community under project no. 26220220180: Building Research Centre "AgroBioTech".

**Conflicts of Interest:** The authors declare no conflict of interest.

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
