# Peer review of "Identification of Yeasts with Mass Spectrometry during Wine Production"

_fermentation, doi:10.3390/fermentation6010005_

Round 1
Reviewer 1 Report
Kačániová et al. investigated yeast biodiversity in different grapes, wines and “federweisser”. The topic is interesting but it is too descriptive. Several criticisms are present. It is a too preliminary study; yeast isolates are not typed and further characterization is lacking. Very obvious data have been obtained, the presence of detected is well known. Strains differentiation would have been more appropriate in order to identify biotypes and highlight the biodiversity.
Minor points
Microorganisms should be reported in italics
Typing errors are present
References are not updated
Statistical analysis to better differentiate the samples
Author Response
The authors would like thank both reviewer for their valuable comments for improvements the manuscript.
Reviewer 1:
Kačániová et al. investigated yeast biodiversity in different grapes, wines and “federweisser”. The topic is interesting but it is too descriptive. Several criticisms are present. It is a too preliminary study; yeast isolates are not typed and further characterization is lacking. Very obvious data have been obtained, the presence of detected is well known. Strains differentiation would have been more appropriate in order to identify biotypes and highlight the biodiversity.
Minor points
Minor points were changed.
Microorganisms should be reported in italics
Microorganisms were reported in italics.
Typing errors are present
Typing errors were corrected.
References are not updated
All references were checked and updated.
Statistical analysis to better differentiate the samples
Data with additional statistical analysis is included.
Reviewer 2 Report
Paper should be improved prior acceptance. Several sentences are unclear or ambiguous. Some parts seem to be unlinked each other. See "Results and Discussion" section, for example.
Material and methods
Please, indicate if the wine collected was fermented with microbial starter
-line 74...how wine may be collected from Vitis vinifera?
pag 55-64 and 161-162, please reports in italics the name of yeast species
Line 66-68, please rewrite, bit unclear
Line 161-162..seem to be out of the contest (please, editing as well)
Line 174-183. Please rewrite, bit unclear. What do you intend as contaminant? Please specify
Line 185-192. Please rewrite in a proper way.
Author Response
The authors would like thank both reviewer for their valuable comments for improvements the manuscript.
Reviewer 2:
Paper should be improved prior acceptance. Several sentences are unclear or ambiguous. Some parts seem to be unlinked each other. See "Results and Discussion" section, for example.
The language and sentences were changed.
Material and methods
Please, indicate if the wine collected was fermented with microbial starter
Microbial starter was added.
-line 74...how wine may be collected from Vitis vinifera?
The sentence was corrected as “grape, federweiser from Vitis vinefera were collected”.
pag 55-64 and 161-162, please reports in italics the name of yeast species
All names of microorganisms were reported in italics.
Line 66-68, please rewrite, bit unclear
The sentence was rewritten.
Line 161-162.. seem to be out of the contest (please, editing as well)
The part was rewritten.
Line 174-183. Please rewrite, bit unclear. What do you intend as contaminant? Please specify
The part was rewritten.
Line 185-192. Please rewrite in a proper way.
The part was rewritten.
Round 2
Reviewer 2 Report
paper has been improved
Author Response
Response to Academic Editor Comments
Point 1: Check the text for English grammatical errors (please consult a colleague with an English mother tongue or someone who is very fluent in English) and small typing errors such as line 28: fererweisser, line 40: Kloecker, ...
English errors were changed with Track Changes function in Microsoft Word.
Point 2: Figure 1, 2 3: higher resolution figures should be provided.
Figure 1,2,3 were changed in different program.
Thank you for your comments.